# Present and Future of IgA Nephropathy and Membranous Nephropathy Immune Monitoring: Insights from Molecular Studies

**DOI:** 10.3390/ijms241713134

**Published:** 2023-08-23

**Authors:** Francesca Zanoni, Matteo Abinti, Mirco Belingheri, Giuseppe Castellano

**Affiliations:** 1Division of Nephrology, Dialysis and Kidney Transplantation, Fondazione IRCCS Ca’ Granda Ospedale Maggiore Policlinico, 20122 Milan, Italy; matteo.abinti@unimi.it (M.A.); mirco.belingheri@policlinico.mi.it (M.B.); giuseppe.castellano@unimi.it (G.C.); 2Department of Clinical Sciences and Community Health, Università degli Studi di Milano, 20122 Milan, Italy

**Keywords:** IgA nephropathy, membranous nephropathy, antibodies, monitoring, precision medicine

## Abstract

IgA Nephropathy (IgAN) and Membranous Nephropathy (MN) are primary immune-mediated glomerular diseases with highly variable prognosis. Current guidelines recommend that greater immunologic activity and worse prognosis should guide towards the best treatment in an individualized approach. Nevertheless, proteinuria and glomerular filtration rate, the current gold standards for prognosis assessment and treatment guidance in primary glomerular diseases, may be altered with chronic damage and nephron scarring, conditions that are not related to immune activity. In recent years, thanks to the development of new molecular technologies, among them genome-wide genotyping, RNA sequencing techniques, and mass spectrometry, we have witnessed an outstanding improvement in understanding the pathogenesis of IgAN and MN. In addition, recent genome-wide association studies have suggested potential targets for immunomodulating agents, stressing the need for the identification of specific biomarkers of immune activity. In this work, we aim to review current evidence and recent progress, including the more recent use of omics techniques, in the identification of potential biomarkers for immune monitoring in IgAN and MN.

## 1. Introduction

IgA nephropathy (IgAN) and membranous nephropathy (MN) are primary glomerular diseases of adulthood, and, together with Minimal Change Disease and focal segmental glomerulosclerosis, account for 20% of renal disease diagnoses and patients represent 20% of the end-stage kidney disease (ESKD) population [1]. 

In the past years, genome-wide association studies (GWAS) and other research efforts have made incredible progress in understanding the pathobiology of primary IgAN and MN. It is widely demonstrated that an aberrant immune system activation is involved in the pathogenesis of IgAN and MN and has variable impacts on the response to therapy and on the progression to ESKD. According to the most recent *KDIGO* guidelines [2], the treatment of primary glomerular diseases often relies on the maximization of supportive care. Available immunosuppressive therapies do not have sufficient proven efficacy and safety to be routinely used in all individuals with a primary glomerular disease diagnosis. Therefore, it is likely that disease management requires an individualized approach. On this basis, there has been growing interest in the development of biomarkers for patient risk stratification and response to immunosuppressive treatment. 

Proteinuria at the time of diagnosis is an established risk factor of progression to ESKD in all primary glomerular diseases, and it is often used to guide treatment approaches and select patients that will likely benefit from an immunosuppressive treatment course [2]. In addition, a significant reduction in proteinuria is a surrogate biomarker of the response to therapy in primary glomerular diseases, and it is FDA-approved for use as treatment goal in clinical trials. Nevertheless, while suggesting an immune activity, proteinuria also reflects glomerular hyperfiltration and downstream tubulo-interstitial chronic damage, which are unlikely to respond to immunosuppressive treatment, limiting its use as a biomarker of immune monitoring.

In recent years, several studies have focused on the identification of markers of immune activity that could be used, in addition to proteinuria and other clinical risk factors, for the diagnosis, prognosis, and treatment selection, as well as prediction and management of post-transplant recurrence. 

In this review, we illustrate the current knowledge on IgAN and MN pathogenesis based on molecular and genomic studies and discuss recent progress on the identification of potential biomarkers of immune activity. In particular, we describe available and novel molecules and their potential use for prognosis assessment, immune monitoring, and therapeutic guidance, with a focus on omics approaches. The criteria used for conducting this narrative review are summarized in Appendix A.

## 2. IgA Nephropathy

IgAN is the most common glomerular disease worldwide, with an estimated frequency of 2.5 cases per year per 100,000 adults [3]. The disease progresses to end-stage kidney disease (ESKD) in 40% of cases over 20 years. The average age of disease onset is 30 years, and clinical symptoms include persistent or occasional macro-hematuria, micro-hematuria, proteinuria, and systemic hypertension [4,5]. The prevalence, presentation, progression, and course of IgAN are highly variable and are especially dependent on age of disease onset [5]. Diagnosis of IgAN requires a kidney biopsy with a demonstration of tissue staining for dominant or co-dominant IgA1 immune complex deposits in the mesangium of the glomerulus, and identifiable mesangial proliferation [6]. This is often accompanied by C3 and with a varying frequency of IgG and IgM deposits [7]. Under light microscopy, pathological lesions range from mild mesangial hypercellularity, to diffuse inflammatory features in which mesangial hypercellularity is seen with the addition of endocapillary leukocyte proliferation and/or crescents. Crescentic forms of IgA nephropathy (with >50% of glomeruli with crescents) are very rare. In late stages, focal segmental glomerulosclerosis and global sclerosis are often evident [8,9]. IgAN prognosis is highly variable, with ESKD risk that varies between 7 and 60%. Some of this variability, which is partially explained by the different approaches in urinary screening and biopsy indication, and by follow-up time differences across studies, probably reflects true heterogeneity in disease activity and prognosis.

## 3. Genetic Susceptibility and Pathobiology

Population-based genetic studies on adult IgAN have given new insights into the pathobiology of the disease, and the discovery of genome-wide significant loci suggested the implication of several pathogenic pathways, demonstrating that IgAN is rather a polygenic disease. There have been six major GWASs conducted since 2010 in populations of adults with IgAN [10,11,12,13,14,15]. The latest published large-scale GWAS, including over 10,000 IgAN cases and nearly 30,000 controls of multiple ancestries, discovered 30 genome-wide significant risk loci, implicating defects in antigen presentation, complement pathway, intestinal IgA production and innate immunity against mucosal pathogens [15]. 

These IgAN risk loci identified have helped to reformulate the multi-hit model of IgAN pathogenesis. A defect in glycosylation on the hinge region of IgA1 leads to the secretion of polymeric Gd-IgA1 molecules (hit 1). This defect is thought to be highly genetically driven and has an estimated heritability of 50–70% [12,16]. An aberrant activation of the mucosal immune system, particularly in the bowel, in response to exogenous antigens may be involved in stimulating Gd-IgA1 overproduction [10,15].

Gd-IgA1 molecules constitute a new epitope that induces the secretion of specific autoantibodies (hit 2). The origin of these antibodies is still poorly understood. Genetic predisposition involving the immune response and intestinal mucosal activity could lead to the production of specific IgG or IgA anti Gd-IgA1 [17,18]. Additionally, molecular mimicry of bacterial antigens could lead to the production of cross-reactive antibodies against Gd-IgA1 [19,20].

Circulating IgG and Gd-IgA1 immune complexes (hit 3) are specifically found in IgAN and are thought to be fundamental for renal damage [21,22,23]. IgG and Gd-IgA1 immune complexes cleave and bind the soluble IgA Fc alpha receptor (CD89), which appears to be important for the pathogenesis of renal damage [24,25,26]. The deposition of immune complexes induces mesangial proliferation through the interaction between CD89, mesangial transferrin receptor 1 and transglutaminase 2 [27]. Such large immune complexes elute hepatic clearance and can be deposited in the mesangium (hit 4). It has been shown that Ig-Gd-IgA1 complexes, but not Gd-IgA1 alone, induce proliferation of mesangial cells and their secretion of extracellular matrix and cytokines such as Tumor Necrosis Factor (TNF) alpha and Interleukin 6 (IL-6) (hit 4). Ultimately, the activation of the alternative and lectin pathway of the complement system contributes to glomerular inflammation and damage in IgAN [28,29,30].

## 4. Immune Monitoring

Recent elucidations on disease pathobiology have redefined IgAN as an autoimmune disease. Nevertheless, immunosuppressive treatment with currently available agents often has not demonstrated enough efficacy and safety to be routinely used in IgAN [2,31].

Findings from the latest GWAS showed that the dysregulation of innate and adaptive immunity involving several different pathways leads to the manifestations of IgAN. These findings indicate that IgAN encompasses a complex group of different disease statuses, where several patterns of immune dysregulation and activity are involved, and each of them may represent a target for novel immunomodulating agents [15]. As new drugs are being developed and tested, there has been a growing need for biomarkers to be used for guidance in treatment selection and disease monitoring.

### 4.1. Available Biomarkers

Established clinical biomarkers for the diagnosis and the prognosis of IgAN are proteinuria, lower eGFR at diagnosis and hypertension [32,33]. Prolonged proteinuria >1 g/day identifies individuals at risk for disease progression for which immunosuppressive treatment could be considered. Nevertheless, persistent proteinuria is often evident in the presence of glomerular hyperfiltration and tubule-interstitial scarring without immune activity. These lesions are unlikely to respond to immune modulating agents and require a different approach. Therefore, while highly informative on the risk of kidney disease progression, proteinuria does not constitute a reliable parameter to account for specific treatment decisions and immune activity monitoring. 

The Oxford IgAN Classification Score stratifies kidney biopsies based on five histological parameters: mesangial hypercellularity (M), endocapillary hypercellularity (E), segmental glomerulosclerosis (S), tubulointerstitial fibrosis/ tubular atrophy (T), and crescents (C), and it has been extensively validated as a biomarker of disease prognosis [34,35,36]. In addition, the IgAN Prediction Tool, which includes the Oxford histological features alongside proteinuria, eGFR, blood pressure, and other clinical parameters, was demonstrated to effectively predict 5-year risk of ESKD in IgAN [37]. Both the Oxford Classification score and the IgAN Prediction Tool are recommended to be routinely used to assess patient risk of the progression of kidney disease by current guidelines [38]. Nevertheless, they are not validated for use to guide disease management and therapeutic approaches.

Recent evidence has shown that each of the histological features in the Oxford Classification Score reflects the activation of specific immune pathways triggered by the mesangial deposition of immunocomplexes. On this basis, several observational studies have focused on the use of each pathological parameter to guide IgAN management. Small retrospective studies have demonstrated that the presence of endocapillary hypercellularity, which reflects macrophage glomerular invasion and immune activation, identifies individuals with a higher likelihood of having a benefit from immunosuppressive treatment [39,40,41]. 

Recent small observations have evaluated the use of repeat kidney biopsy as a tool for monitoring the response to immunosuppressive treatment [41]. While highly informative for IgAN diagnosis and prognosis, kidney biopsy is an invasive procedure associated with morbidity and discomfort, and unlikely to be repeated during the course of the disease. Therefore, its use is often limited for prognosis assessment at the time of diagnosis. 

In summary, while the Oxford Classification score and the IgAN Prediction Tool are currently the best markers for prognosis risk assessment, singular histologic features, in particular the E score, seem to be better associated with immune activity, and, therefore, have the potential to be additionally useful for therapeutic guidance. Randomized clinical trials will be needed to validate such evidence [42].

### 4.2. Novel Molecular Biomarkers

#### 4.2.1. Antibodies and Complement

Based on the four-hit pathogenic model, several research studies have focused on the identification of blood and urinary biomarkers of immune activity in IgAN (Figure 1).

High levels of Gd-IgA1 in circulation (hit 1) have been associated with a higher risk of developing IgAN and may have a negative prognostic role in IgAN-affected individuals in several observational studies [4,43,44,45]. In addition, high pre-transplant serum gd-IgA1 may be predictive of post-transplant IgAN recurrence [46]. Similarly, urinary levels of gd-IgA1 are significantly higher in IgAN than in healthy controls and have been associated with higher levels of proteinuria [47]. Nevertheless, high gd-IgA1 levels have also been found in unaffected family members and in individuals with other glomerular diseases and gut disorders [16,44] and a recent meta-analysis did not confirm its role as an IgAN biomarker [48]. Recent studies have proposed the use of gd-IgA1/C3 plasma levels for the identification of IgAN cases and disease prognosis. Higher levels of gd-IgA1/C3 have been associated with more severe histological lesions, persistent proteinuria, and hematuria, and with the risk of kidney disease progression in a Chinese study [49]. 

Some reports demonstrated that serum levels of gd-IgA1–specific IgG and IgA autoantibodies (hit 2) were higher in IgAN compared with healthy subjects and diseased controls and were associated with IgAN progression and post-transplant recurrence, although other studies demonstrated that these molecules were also high in a portion of CKD individuals without IgAN [22,50].

Recent reports have identified that IgG-gd-IgA1 immune complexes (hit 3) are highly specific for IgAN and could represent a new disease biomarker, as shown in a small study [51,52]. Higher plasma levels of IgA bound to soluble CD89, which is thought to play a major role in immune complex mesangial deposition and glomerular damage, were found in IgAN individuals in a small French study, but the results were not replicated in another European study [53,54]. Vuong et al. demonstrated that a low serum CD89-IgA complex at time of biopsy was associated with a worse prognosis, but this result was not replicated in a Korean cohort [55,56]. Another small study found lower urinary levels of soluble CD89 in individuals with active IgAN and IgA vasculitis, compared with silent disease [57]. Similarly, lower serum CD89-IgA complexes were associated with higher risk of post-transplant IgAN recurrence, as suggested in a small study [58].

Markers of alternate and lectin complement activation (hit 4) in the renal tissue have currently been studied to predict disease progression and response to novel complement-targeting therapies in clinical trials. Small studies have demonstrated that lower circulating levels and glomerular deposition of C3 and CFHR5, a molecule involved in the regulation of the alternate complement pathway, and markers of lectin complement activation (such as MASP, mannose-binding lectin and C4d) were associated with more severe histological lesions, proteinuria, and disease progression [30,59,60,61,62]. 

Altogether, these findings suggest that, while gd-IgA1 and their antibodies levels may not be specific of IgAN, and, therefore, not useful as single biomarkers for IgAN diagnosis, gd-IgA1-Ig and soluble CD89-IgA immune complexes seem to be highly specific and correlated with IgAN immune activity and may develop a role for immune monitoring and therapeutic guidance. Nevertheless, despite the efforts, none of these molecules have proven to be effective for clinical use to date, since their measures often lack standardization across laboratories, and they have been assessed mostly in small studies with inconsistent findings across different populations. 

Future clinical trials will assess the use of complement activation molecules as biomarkers for guidance in targeted therapy indication. 

#### 4.2.2. Immune Cell Subpopulations

GWAS studies demonstrated a prominent role of extra-renal immune cells, such as B and T cells, neutrophils and monocytes, as potential causal cells in IgAN [10,15]. Accordingly, research efforts have been focused on the identification of immune cell subpopulations in the circulation to guide diagnosis, prognosis, and disease management. Few studies have shown higher levels of circulating CXCR5+CD4+ T cells, CD23+ and CD19+CD5+ B cells, an expansion of the non-classical monocyte subset, and the increased expression of CD62L in IgAN [63,64,65,66,67], but none have been found to be specific to IgAN and useful for prognosis assessment and immune monitoring to date.

A more promising field is the study of immune cell profiles at kidney biopsy. Growing evidence suggests a role of CD206+ macrophage infiltration as a biomarker, as recently proposed in a Chinese study: CD206+ macrophages predicted the response to immunosuppressive therapy in individuals with IgAN and histopathologic lesions of a high risk of progression (M1 and/or E1) [68]. Future larger studies will be needed to validate this potential biomarker across different populations and to assess its use to guide treatment management.

## 5. Future Prospects: Omics

Findings from the latest GWAS have raised the hypothesis that IgAN may rather constitute a complex group of different immune-mediated patterns [15]. On this basis, it is highly likely that a single disease biomarker would fail to prove high sensitivity, specificity and predictive value for the diagnosis, prognosis, and management in IgAN. Omics techniques have been extensively used in several diseases, including IgAN, and have contributed to the definition of its pathogenesis. Using integrated data from genomics, epigenomics, transcriptomics and proteomics could define a molecular profile of each IgAN patient, and ultimately may be able to guide the clinician towards a better prognosis assessment and individualized targeted therapy, an approach that is already in use in the field of oncology. Table 1 illustrates potential biomarkers identified through omics studies.

### 5.1. Genomics

Polygenic risk scores (PRS) leverage GWAS results to capture the cumulative effect of genetic variants on disease risk, and, therefore, constitute a quantitative measure of genetic risk [46,69]. The latest published IgAN GWAS to date, which included 10,146 kidney-biopsy-diagnosed cases of European and East Asian ancestries and 28,751 controls, led to the discovery of 30 IgAN risk loci with pleiotropic effects that cumulatively express 11% of disease risk. In addition, this study demonstrated that the overall IgAN variance explained by common single-nucleotide polymorphisms (SNP) is 23%, suggesting a strong heritability [15].

In addition, a genome-wide PRS constructed from the latest GWAS results was associated with a lower age at diagnosis and a higher lifetime risk of ESKD, suggesting that genetic background may be predictive of a more aggressive disease [15]. In addition, an earlier age at diagnosis and at ESKD are known risk factors for IgAN recurrence after transplantation; therefore, higher polygenic risk may be associated with a higher recurrence risk and ultimately graft failure [70]. Although future validations are needed, the use of PRS has the potential to stratify index cases and their family members and guide the clinician towards better disease management. Similarly, the polygenic risk assessment of ESKD individuals and their candidate donors, especially if within a family, has the potential to identify individuals at risk of post-transplant disease recurrence and guide donation and transplant management [70].

In addition, as new therapies are being developed and clinical trials designed, the genomic profiling of an affected individual (such as the identification of the carrier status of IgAN risk alleles), might guide clinicians towards the best therapeutic approach and enrollment in specific clinical trials. 

In summary, the highly polygenic nature of IgAN demonstrated by GWAS studies suggests that PRS may be an additional promising tool for IgAN diagnosis and prognosis assessment. In addition, given that most of the IgAN risk loci map to genes involved in the regulation of immune pathways, the PRS and/or the identification of specific risk alleles mapping to potential druggable pathways may be useful for the identification of individuals that would likely benefit from potential new targeted therapies. Although promising, the use of genomics in IgAN management will need to be assessed in large-scale validation studies involving different populations and in clinical trials.

### 5.2. Transcriptomics

Transcriptomic analysis relies on recently developed techniques of RNA isolation in several tissues. In recent years, bulk, single-cell, single-nuclei, and RNA sequencing, and assay for transposase-accessible chromatin with sequencing (ATAC-seq) techniques have also been developed in the setting of glomerular disease and are being used for the identification of specific cells in various compartments of the kidney and their expression profiles. Their use has helped improve knowledge on the molecular pathogenesis of IgAN and has identified potential pathogenic biomarkers and druggable pathways.

A pilot study using bulk RNA-sequenced kidney biopsies identified specific differentially expressed genes in IgAN with endocapillary proliferation, compared with no endocapillary proliferation, and suggested their use for therapeutic guidance [71]. Reich et al. discovered differentially expressed genes in the kidney tissues of IgAN, compared with healthy controls, that are molecular signatures of proteinuria and may be used as prognostic biomarkers [72].

Single-cell RNA sequencing studies have identified several differentially expressed genes in mesangial cells, resident macrophages and CD8+ T cells, as well as tubular cells [73,74,75]. Many of these genes are involved in immune, inflammatory and proliferation pathways and have helped to define the molecular pathogenesis of IgAN. 

Although impacted by a small sample size and lack of validation, these results suggest that the use of a kidney expression profile at the time of biopsy may identify cases with a higher risk of progression that could benefit from future targeted immunomodulating agents.

### 5.3. Epigenomics

Epigenomics is the study of changes in DNA availability and, therefore, the regulation of gene expression. One small genome-wide epigenomic study in IgAN identified two hypomethylated regions and one hypermethylated region in genes involved in T-cell receptor signaling, that may promote a shift towards a T helper 1 immune response in IgAN [76].

MicroRNAs (miRNAs) are short oligonucleotides that regulate gene expression by disrupting translation. Small pilot studies have discovered that several miRNAs detected in kidney biopsies or in circulation may have a prominent pathogenic role in IgAN: miRNAs 148b, 374b and let-b may regulate gd-IgA1 production, miRNAs 877-3p and 100-3p may be involved in the stimulation of mesangial cells by gd-IgA1, miRNAs 21-5p, 155, 199a-5p, 205, and 214-3p with glomerulosclerosis and interstitial fibrosis and kidney function decline [77,78,79,80,81,82]. Few reports show changes in urinary miRNA excretion in IgAN.

Although many of the studies lack validation of the results, there is increasing interest in considering miRNAs as potential useful biomarkers for disease prognosis and monitoring, and an important source for the development of new targeted therapies. Future studies will need to assess the use of miRNAs for clinical guidance.

### 5.4. Proteomics

Proteomics, the study of blood or urinary protein=-level patterns, has helped understanding the pathobiology of IgAN. Small studies have identified urinary proteins of biological plausibility that seem to be specifically expressed in IgAN patients compared to diseased and healthy controls [83,84,85]. Many of these are complement system components, coagulation factors, intracellular, and transmembrane proteins, and molecules involved in oxidative stress, and may be associated with a worse prognosis and response to therapy [85,86,87,88,89].

In conclusion, several different molecules have proven to be associated with IgAN risk and prognosis. However, future studies will be needed to better investigate the use of proteomics for immune profiling and its role in management guidance.

**Table 1 ijms-24-13134-t001:** Potential biomarkers for IgAN immune monitoring identified through omics techniques.

Biomarker	Current Evidence	Role in Disease Monitoring	Ref
GENOMICS			
IgAN PRS	Associated with lower age at diagnosis and lifetime risk of ESKD	It may be indicative of worse prognosis	
		It may be useful for risk stratification of native IgAN, transplant candidates and potential donors	[10,15]
IgAN risk loci	- Associated with higher risk of IgAN - Potential new druggable targets	Carrier status of specific IgAN risk loci (i.e., risk allele at *CFH* locus) may guide towards therapeutic approach and enrollment in clinical trials (i.e., for anti-complement therapies)	[15]
TRANSCRIPTOMICS			
A total of 424 differentially expressed genes at bulk RNA-seq in IgAN glomeruli with and without endocapillary proliferation (E1 vs. E0): - *C1QA, C1QB, C2, VSIG4* (innate immune response and classical pathway complement activation genes) - *HSPE, TIMP1* (matrix degradation and turnover genes) - CD163 (M2 macrophage polarization gene)	- Differentially expressed genes in 22 IgAN patients with E1 vs. E0 - Correlated with eGFR at time of biopsy	They may be used for therapeutic guidance	[72]
A total of 11 differentially expressed proteinuria-associated genes in IgAN vs. healthy controls: *COL1A1, ELF3, EGR1, IER3, HBEGF, HBEGF, MAFF, MCL1, SAMD4A, SERPINE1, STEAP1, TYMS* (involved in regulation of proliferation and differentiation of immune and epithelial cells)	- Differentially expressed in IgAN vs. healthy controls - Correlated with proteinuria in IgAN	They may be used as prognostic biomarkers	[73]
Differentially expressed genes at single-cell RNA-seq in kidney tissue	Upregulation of *JCHAIN* expression in IgAN mesangial cells	It may constitute a new molecular druggable target	[74]
	Upregulation of proinflammatory genes in kidney resident macrophages, downregulation of cytotoxic marker genes in CD8+ T cells in IgAN	Potential biomarkers of IgAN diagnosis, potential new drug targets	[74,75]
	Differentially expressed genes in endothelial cells and tubular cells in IgAN	Potential biomarkers of IgAN diagnosis, potential new drug targets	[76]
EPIGENOMICS			
miRNAs 148b, 374b and let-7b	Associated with regulation of gd-IgA1 production	Potential new biomarkers of IgAN diagnosis and druggable targets	[78,79,80]
miRNAs 877-3p and 100-3p	Associated with overproduction of IL-8 and IL-1beta in mesangial cells	Potential new biomarkers of IgAN diagnosis and druggable targets	[81,82]
miRNAs 21-5p, 155, 199a-5p, 205, and 214-3p	Associated with fibrosis and kidney function decline	They may have a negative prognostic role	[83]
PROTEOMICS			
Differential urinary concentrations of - afamin, - leucine-rich alpha-2-glycoprotein, - ceruloplasmin, - alpha-1-microgolbulin, - hemopexin, - apolipoprotein A-I, - complement C3, - vitamin D-binding protein, - beta-2-microglobulin, - retinol-binding protein 4	Correlation with histological scoring system, especially endocapillary proliferation	They may be used as prognostic biomarkers	[84,85,86,87,89]
Differential urinary concentrations of - kininogen, - inter-alpha-trypsin-inhibitor heavy chain 4 (35 kDa fragment), - transthyretin	Correlation with response to ACEi therapy	Potential predictive role to ACEi response	[86,90]

Abbreviations: IgAN: IgA Nephropathy; PRS: polygenic risk score; ESKD: end-stage kidney disease; *CFH*: complement factor H; eGFR: estimated glomerular filtration rate; miRNA: micro-RNA.

## 6. Membranous Nephropathy

Primary MN is a rare glomerular disease, with an estimated annual incidence of 1.2 cases in 100,000 adults [3]. The typical clinical symptom is nephrotic syndrome, often characterized by the slow progression of edema over weeks or months. Microhematuria can occur in 30% of cases and hypertension in 10% [90]. Although it is a rare disorder, it is a major cause of nephrotic syndrome in adults of European ancestry, and it accounts for over 20% of nephrotic syndrome cases overall [91]. While it can present at any age in all ethnic groups, the incidence peak is between 30 and 50 years of age [92]. 

Histological diagnosis of MN relies on immune-fluorescence and electronic microscopy, with the detection of diffuse immune deposits in the sub-epithelial compartment of the glomerular basement membrane (GBM). Light microscopy features can range from minimal or no glomerular changes to progressive thickening of the GBM and subsequent reabsorption of the immune deposits, a pattern shared among both primary and secondary forms of MN. In primary MN, the immune fluorescence specifically shows diffuse granular deposits of IgG (mostly IgG4) with or without C3 [91,93]. The natural history of MN is characterized by either a spontaneous remission of nephrotic syndrome (in ~30% of cases), or a stable or progressive disease course with deterioration in renal function (in ~30% and ~30% of cases, respectively). In these latter cases, if left untreated, or if no remission occurs with therapy, its natural history is characterized by progressive renal failure with ESKD occurring in 50% in 10–15 years [92].

## 7. Genetic Susceptibility and Pathobiology

While secondary forms of MN are characterized by nephrotic syndrome because of a specific insult (such as infections, malignancy, systemic immune-mediated diseases, and drug toxicity) [94,95], primary MN is a non-inflammatory autoimmune disorder that belongs to the category of podocytopathies, and it is caused by circulating pathogenic autoantibodies against podocyte antigens.

The discovery of antibodies against the phospholipase A2 receptor 1 (PLA2R), in 60–70% of MN cases, or thrombospondin type-1 domain-containing 7A (THSD7A), in 5% of anti-PLA2R negative cases, has revolutionized the knowledge of primary MN pathogenesis and its management [93,96].

Early genetic studies in European populations found that MN was associated with certain *HLA* antigens and with a locus located within the *PLA2R1* gene [97,98,99]. Recent multi-ethnic GWAS performed on nearly 13,000 individuals of East Asian and European ancestries discovered six independent loci associated with MN risk: three located within the *HLA* region, one in the *PLA2R1* locus and two new genome-wide significant loci, in *NFKB1* and *IRF4* [100]. These findings confirm the autoimmune nature of the disease and demonstrate the pathogenic role of antibodies against PLA2R. As suggested by these studies, genetic predisposition with a permissive *HLA* haplotype might allow for higher or altered immunogenicity or expression levels of podocyte PLA2R and enhance the production of anti-PLA2R autoantibodies [101,102,103]. 

PLA2R and THSD7A are transmembrane proteins expressed in the podocytes of normal human glomeruli. They bind podocytes’ transmembrane proteins and co-localize with IgG4 within sub-epithelial deposits, activating the alternative complement pathway with the formation of the membrane-attach complex (MAC, or C5b-9) [104]. The deposition of MAC alters several podocyte metabolic pathways, including the upregulation of local oxidative stress, the structure and function of its cytoskeleton and the expression of slit diaphragm proteins [105,106,107]. The endocytosis of C5b-9 and the production of misfolded proteins is responsible for podocyte endoplasmic reticulum stress and, consequently, apoptosis [108,109,110,111]. Moreover, injured podocytes produce new extracellular matrix around the immune deposits, responsible for the progressive thickening of the GBM. Autoantibodies also may be involved in podocyte changes, through the alterations of transmembrane signals of their specific antigens [112,113].

## 8. Immune Monitoring

Treatment of MN includes supportive therapy for the management of nephrotic syndrome, and immunosuppression. Alkylating agents combined with steroids have long been the cornerstone of MN treatment, alongside calcineurin inhibitors, often used as a second-line or alternative agent [90].

After the discovery of circulating autoantibodies, research trials have focused on anti-B cell therapy, and rituximab has been demonstrated to be efficacious in a significant proportion of patients [95,114,115,116]. Despite that, it is still debated who would benefit from immunosuppressive therapy, the timeline for its initiation, or the type of drug to use. In this setting, biomarkers of immune activity have a crucial role for the identification of individuals at risk of progression and help personalize treatment to avoid useless exposure to the many side effects of immunosuppressive medication.

### 8.1. Available Biomarkers

According to the most recent KDIGO guidelines [2], gold standards for risk stratification in pMN are proteinuria and eGFR. Higher proteinuria and lower eGFR identify patients at higher risk of progression, in which immunosuppressive treatment should be considered. Specific urine protein concentrations are thought to be of additional aid in patients risk stratification, such as alpha 1 microglobulin, IgG, and beta 2 microglobulin, as well as a high selectivity index, calculated as ratio between clearance of IgG and clearance of albumin [2].

Nevertheless, persistent or increased proteinuria after treatment may be caused by hyperfiltration and irreversible capillary wall injury, that will unlikely respond to immunosuppressive therapy. Therefore, while proteinuria is a biomarker of glomerular membrane impairment, it does not reflect the pathogenic mechanism behind it, and does not strictly correlate with the immunological activity of the disease.

#### 8.1.1. Anti PLA2R Antibodies

Anti-PLA2R are detectable in around 80% of patients with MN and their levels correlate with prognosis and clinical outcome. According to KDIGO guidelines, individuals with nephrotic syndrome and positive anti-PLA2R antibodies may not need renal biopsy because the sensitivity and specificity of blood PLA2R antibody levels for the diagnosis of primary MN in the active phase are around 74% and 95%. 

The formation and deposition of immune complexes in primary MN anticipate the clinical manifestations of the disease. They also correlate with the clinical course of the disease, so that higher serum anti-PLA2R are associated with higher proteinuria and creatinine at presentation, as well as a lower likelihood of spontaneous or pharmacological remission [117,118]. Anti-PLA2R antibody decline also anticipates proteinuria decline, and increased titers predict and anticipate relapse, suggesting that they are a marker of the immunological activity of MN [119]. Persistent positivity after treatment is predictive of clinical resistance to immunosuppressive therapy [120]. 

Recent studies suggested that anti PLA2R levels correlate with the degree of epitope spreading [121]. Epitope spreading occurs in several autoimmune diseases to expand the antibody repertoire and increase the overall immune response. Epitope spreading for the CysR epitope of PLA2R was found to be an independent predictor of reduced renal survival in MN and the absence of epitope spreading at disease onset predicted remission. Moreover, in a recent study, among 17 patients with epitope spreading at baseline, treatment with Rituximab reversed epitope spreading at 6 months [122]. 

Additionally, recurrent MN occurs in 30% to 50% of people with kidney transplantation (KTx) after ESRD. Higher levels of anti-PLA2R antibodies are associated with frequent recurrence after KTx. To evaluate therapy response and modify the treatment strategy, KDIGO 2021 recommended that anti-PLA2R antibody levels should be monitored longitudinally every 3 or 6 months following KTx [2].

PLA2R can be detected in the subepithelial compartment of the GBM in individuals with MN, and it correlates with positivity of anti PLA2R in the circulation [123]. Nevertheless, since glomerular PLA2R can be detected in absence of anti PLA2R at either early stages of immune activity or in a no longer active disease, it is not considered a reliable biomarker for immune monitoring in MN [124].

In summary, the prognostic and predictive role of anti-PLA2R antibodies demonstrated in the aforementioned studies suggests that its dosage could be used to personalize the management of MN patients and tailor treatment to reduce side effects related to excessive immunosuppression. However, while anti-PLA2R is a biomarker of humoral response in MN, it does not fully capture the immunological activity of the disease [125]. For example, memory B cells may be able to develop a sustained immune response in the absence of antibodies in circulation. This suggests that other methods that fully assess the immunological activity of MN may improve the understanding of the disease pathobiology and help monitor immune response. Table 2 illustrates new potential biomarkers for MN clinical monitoring.

#### 8.1.2. Anti-THS7DA

As mentioned, studies have demonstrated the pathogenic role of anti-THSD7A in MN. Anti-THSD7A has a high diagnostic value for anti-PLA2R-negative MN and can be used as an auxiliary diagnostic method. Its total sensitivity for the diagnosis of MN was found to be 4%, while the specificity was 99% [126]. However, anti-THSD7A antibody is unable to fully discriminate MN from secondary forms [2]. Therefore, due to the high prevalence of cancer in MN patients with anti-THSD7A antibody positivity, it is advised to undergo a malignancy screening [126].

Recent research has demonstrated a correlation between the anti-THSD7A antibody levels and both disease activity and therapy response. Higher anti-THSD7A titers predicted worse prognosis [127]. As a result, monitoring anti-THSD7A levels could guide towards treatment decisions and disease management [128]. Unfortunately, no specific treatments have been investigated in these patients due to the rarity of this condition [129].

### 8.2. Additional Target Antigens and Antibodies in MN

Patients with MN who tested negative for antibodies against PLA2R1 and THSD7A were recently found to have circulating antibodies against serine protease HTRA1 [130], Netrin G1 (NTNG1) [131] and Contactin 1 (CNTN1) [132]. These proteins are produced by podocytes and co-localize in glomerular immune deposits with IgG4 antibodies. Anti-CNTN1 antibodies have been found in individuals with chronic inflammatory demyelinating polyneuropathy (CIDP), which raises the possibility that the autoimmune response in MN and the pathophysiology of the underlying neurological lesion are linked [131]. These three antigens still lack clinical relevance and an established illness onset mechanism. Nonetheless, autoantibody levels and proteinuria were tightly related in a small number of patients with clinical follow-up [133].

Other possible non-podocitary antigens include Exotoxin 1 (EXT1) [134], Exotoxin 2 (EXT2) [134], Neural Epidermal Growth Factor-Like 1 (NELL1) [135], Semaphorin 3B (SEMA3B) [136], Protocadherin 7 (PCDH7) [137] and Neural cell adhesion molecule 1 NCAM1 [138]. The interaction of these molecules with immunoglobulins in subepithelial immune deposits and the identification of circulating antibodies against several of these antigens in MN patients provide evidence for their possible involvement as MN antigens [139]. Malignant tumors might be present in anti-NELL-1 positive MN. About 11.7% to 33% of MN with malignant tumors are associated with anti-NELL-1. MN can sometimes occur before malignant tumors are found. MN patients who have NELL-1 serum antibody positivity, for this reason, may need regular follow-ups [135]. MN linked to SEMA3B can affect children and young adults [136]. NCAM1 seems to be strongly related to a subset of patients with membranous lupus nephritis [138].

In summary, it seems premature to assume that all MN antigens and antibodies will have the same clinical relevance regarding differential diagnosis, treatment management, and prognosis prediction as PLA2R. A deeper comprehension of antigen-specific pathophysiology and the use of antigen-specific animal models will be necessary to evaluate novel treatment strategies for MN.

### 8.3. Immune Cells

Immune cells play a role in the onset and progression of MN. Increases in the CD4+/CD8+ T-cell ratio and Th2/Th1 cell ratio have been observed in the peripheral blood of MN patients. This rise induces the production of IgG4 by B cells, under the stimulus of IL-4 and IL-10 [140]. Rituximab, an anti-CD20 monoclonal antibody, has positive clinical outcomes in the treatment of MN. However, peripheral blood B cells are not an accurate biomarker of disease activity and response to therapy to date [141]. 

Certain T cell and monocyte subsets have recently been proposed as potential biomarkers to assess the progression and outcome of patients with MN. According to Motavalli et al., in MN there is a reduced expression of Foxp3 and a lower percentage of Treg cells in the peripheral blood [142]. Yet, in response to rituximab, Treg cells greatly rise. In patients who had a clinical response to rituximab, the percentage of Tregs was lower at baseline and considerably higher on day 8 of therapy, while it remained unchanged in non-responders [142,143]. The creation and operation of the germinal center are greatly dependent on Tfh cells, a CD4+ T lymphocyte subgroup that cause, through IL-21, plasma cell differentiation and immunoglobulin production [144]. Shi et al. discovered a favorable correlation between 24 h urinary protein levels and ICOS+ and PD-1+ Tfh cells in peripheral blood of individuals with MN [145]. Human PBMCs can differentiate into CD14+, CD163+, CD206+, and CD115+ M2 monocytes that release anti-inflammatory mediators, as interleukin IL-10. According to Hou et al., there was a significant correlation between the quantity of these M2-like cells, 24 h urine albumin levels and serum PLA2R in individuals with early MN [146].

In summary, specific T cell and monocyte subpopulations have shown associations with higher immunological activity in MN. However, current evidence does not support their role in the clinical setting, stressing the need for future larger validation studies.

**Table 2 ijms-24-13134-t002:** New potential MN biomarkers and their role in clinical monitoring according to current evidence.

Biomarkers	General Characteristics	At Disease Onset	For Disease Monitoring and Prognosis	Ref.
Circulant Antibodies against podocitary antigens				
Anti-THS7DA	- Useful in patients tested negative for anti-PLA2R - High diagnostic value for primary MN (Sn of 4%, Sp of 99%) - high cancer prevalence in anti- THS7DA positive MN	- Correlation between anti-THS7DA and disease activity at disease onset - At onset, malignancy screening is needed - To date, no specific treatment has been tested for anti-THS7DA positive MN	- Monitoring Ab levels could guide treatment decisions and disease management - Neoplastic monitoring is recommended in these cases - Higher Ab levels predict worse prognosis	[126,127,128]
Anti-HTRA1 Anti-NTNG1 Anti-CNTN1	- Useful in patients tested negative for anti PLA2R and THSD7A - Overlap with chronic inflammatory demyelinating polyneuropathy	- A tight relation of Ab levels with proteinuria, seen in small number of patients - Pathogenetic mechanism is still not defined	- No data on monitoring Ab levels - No data on their impact on prognosis	[130,131,132]
Circulant Antibodies against non-podocitary antigens				
Anti-EXT1 Anti-EXT2 Anti-NELL1 Anti-SEMA3B Anti-PCDH7 Anti-NCAM1	- About 20% of MN with malignant tumors are associated with anti-NELL1 positivity - Anti-SEMA3B positive MN is more frequent in children and young adults - Anti-NCAM1 positive MN may be related to membranous lupus nephritis	- Assess anti-SEMA3B in childhood and in young adults with nephrotic syndrome - Assess anti-NCAM1 in membranous lupus nephritis - At onset, malignancy screening is needed, particularly if anti-NELL1 positivity	- No data on monitoring Ab levels - No data on their impact on prognosis	[134,135,136,137,138]
Immune Cells				
CD4+/CD8+ T-cell ratio Th2/Th1 cell ratio CD20 B cells Treg cells ICOS+, PD-1+ Tfh cells CD14+ CD163+ CD206+ CD115+ M2 monocytes	- Immune cells are involved in the onset and progression of MN - B cells are the target of Rituximab, but their levels do not reflect disease activity and response to therapy - Novel immune biomarkers are needed, to clarify the pathogenesis and the progression of the disease	- An increase in these ratios is observed in in the peripheral blood of MN patients - A lower percentage of Treg cells is observed (due to a reduced expression of Foxp3) - There is a strong correlation between Tfh cells and 24 h proteinuria - M2 monocytes correlate with 24 h proteinuria and serum anti-PLA2R in early MN	- After Rituximab, Treg concentrations increase in responder patients - Their levels in peripheral blood may be influenced by the disease activity, sCr and uProt, and may be used for immune monitoring - No data on their impact on prognosis	[140,141,142,143,145,146]
**Genomics**				
6 MN-risk loci	- PRS may be useful when a renal biopsy is not available or inconvenient - In KTx settings, donor risk alleles for MN predict post-transplant recurrence	- PRS is associated with higher proteinuria and anti-PLA2R positivity	- PRS may be useful for risk stratification and prognosis - PRS could be useful to stratify transplant candidates and their donors, to reduce recurrence risk	[100]
Transcriptomics				
miRNA-130a-5p miRNA-217 miRNA-193a miRNA-186 miRNA-106a, -19b, and -17 miRNA-107, -423-5p lncRNA XIST circ101319 circ0000524	- Patients with MN have different amounts of non-coding RNA compared to healthy controls - They seem to have a pathogenetic and prognostic value - They influence podocyte death, acting on Angiotensin II pathway and cell cycle (i.e., WT1/PODXL pathway)	- Correlation with proteinuria and total cholesterol during the active period of the disease - They are a potential therapeutic target, but no specific data are available	- Correlation with severity and poor outcome in small number of patients - No data about monitoring their expression levels over time	[147,148,149,150,151,152,153,154,155]
Proteomics and Metabolomics				
cationic and acidic albumin SAA1 MIF	- Identification of novel potential biomarkers using specific MS techniques	- Correlations with clinical and pathological features are still poorly understood - cationic and acidic albumin could be used to distinguish primary from secondary MN	- SAA1 seems to predict clinical response to CNi - MIF could be useful to predict response to immunosuppression - No data about monitoring Ab levels over time	[156,157,158]

Abbreviations: MN: Membranous Nephropathy; Sn: sensitivity; Sp: specificity; Ab: autoantibodies; PRS: polygenic risk score; KTx: kidney transplantation; miRNA: micro-RNA; MS: mass spectrometry; CNi: calcineurin inhibitors.

## 9. Future Prospects: Omics

### 9.1. Genomics

Recent GWAS has suggested that MN has an oligogenic architecture where few common risk loci carry most of the genetic risk. Herein, the six MN-risk loci exert unusually high odds of disease risk, and they cumulatively explain 30% of disease variance, while the overall SNP-based heritability was 43% in individuals with East Asian ancestries and 36% in Europeans [100].

A PRS, expressed as the sum of the weighted effects of each of the six risk variants, was demonstrated to explain 32% and 25% of disease risk in East Asians and Europeans, respectively. In addition, when added together with the serologic testing for anti-PLA2R, the PRS increased the overall sensitivity and improved MN prediction, with an area under the Receiver Operating Characteristics (AUROC) curve of over 90% in an external validation cohort. In anti-PLA2R-negative cases, the PRS was demonstrated to correctly re-classify between 20 and 37% of MN cases, suggesting that PRS may be a complementary diagnostic tool, especially in cases in which a renal biopsy is not available or inconvenient from a risk/benefit perspective [100]. Moreover, the PRS was associated with higher proteinuria and anti-PLA2R positivity at disease onset, two independent risk factors for MN progression to ESKD, suggesting that the PRS may be useful for risk stratification and prognosis.

In the KTx setting, a recent report demonstrated that donor MN risk alleles significantly predicted risk of post-transplant MN recurrence [159]. These findings suggest that the use of PRS may be extended to stratify transplant candidates and their donors for the assessment of post-transplant risks [46,69].

In conclusion, genetic risk assessment through the calculation of a PRS may be used as an additional tool for MN diagnosis, while larger studies will need to assess its role in ESKD and post-transplant MN recurrence prediction.

### 9.2. Transcriptomics

Recent research has revealed that patients with MN have different amounts of non-coding RNA, such as miRNA, expression in urine, blood, and kidneys compared with healthy controls. 

To date, aberrant miRNAs seem to be involved in several pathogenic processes in MN pathogenesis and may predict prognosis. For example, Liu et al. demonstrated that lower levels of miRNA-130a-5p in renal biopsy specimens seem to accelerate podocyte death by upregulating PLA2R expression [147]. According to Li et al., miRNA-217 expression in renal tissue was downregulated in patients with MN. Its upregulation can prevent the production of TNFSF11 and protect podocytes from damage [148]. Another study showed that MN patients had increased urinary levels of miRNA-193a and decreased WT1/PODXL expression compared to healthy controls [149]. Sha et al. demonstrated that miRNA-186 seems to be significantly downregulated in the renal tissue of patients with MN, and the use of a miRNA-186 mimic reversed the effects of apoptosis in podocytes caused by Angiotensin II [150]. As observed by Wu et al., the serum levels of miRNA-106a, miRNA-19b, and miRNA-17 in MN patients were significantly lower than in the control healthy group [151]. Recent evidence determined other miRNAs in renal biopsies associated with MN, such as miRNA-107 and miRNA-423-5p, that may control pathways involved in cell cycle, and their levels negatively correlate with proteinuria and total cholesterol [152]. 

Additionally, the expression of the lncRNA XIST was positively correlated with MN severity in a small study [153]. Jin et al. found that the expression of circ101319 was dramatically increased and successfully predicted MN [154]. Circ0000524, which was highly elevated in MN kidney biopsies, by regulating miRNA-500a-5p sponging and CXCL16 expression, seemed to decrease angiotensin II-induced podocyte death [155].

While further validation is needed, altogether these studies suggest that miRNAs may be used as complementary markers of MN diagnosis and prognosis and may constitute targets for novel therapeutic agents.

### 9.3. Proteomics

Proteomic and metabolomic analysis through Mass Spectrometry technology led to the discovery of new potential biomarkers in primary MN. Higher urinary levels of transmembrane and secreted proteins, as well as volatile organic chemicals, have been found in patients affected by MN compared with healthy or diseased controls [160,161,162,163]. A recent study reported that cationic and acidic albumin may be utilized to differentiate primary MN from secondary forms thanks to a recently established technique named capillary isoelectric focusing–mass spectrometry (CIEF-MS) [156]. Yu et al. screened distinct proteins between an IMN remission group and non-remission group following treatment with calcineurin inhibitors (CNi), using nano-HPLC-MS. In the remission group the amount of serum amyloid A1 protein (SAA1) was considerably greater, as if it might predict whether calcineurin inhibitor treatment will be effective for MN [157]. Lastly, using matrix-assisted laser desorption/ionization mass spectrometry imaging (MALDI-MSI), a macrophage migration inhibitory factor (MIF) was recently proposed to predict MN patients’ responses to immunosuppression, comparing two groups of patients with different responses to the Ponticelli protocol [158].

In summary, the identification of these novel biomarkers offers new insights for diagnosing and monitoring MN, although their validation and correlation with clinical manifestations and pathological features are still poorly understood.

## 10. Conclusions

In summary, given the high complexity of the immune pathways involved in the pathobiology of IgAN, and despite recent advancements, there is still lack of reliable biomarkers for immune monitoring. The Oxford IgAN classification and the IgAN Prediction Tool are recommended for use as prognostic markers but have not demonstrated a specific role in monitoring immune activity. New insights from omics studies have suggested that a complex group of different immune-mediated pathways lead to the manifestations that define IgAN, expressing the need for a precision medicine approach. Genomics, epigenomics, transcriptomics, and proteomics are becoming more available and less expensive, and represent unbiased methods to investigate, overall, the immune profile of IgAN. Nevertheless, they generate large amounts of data that are difficult to interpret in most instances. Multi-omics approaches, by integrating information obtained from such techniques, and with the use of machine learning methods, may be able to better elucidate the immune mechanisms involved in the development of IgAN, and ultimately may be used for deep phenotyping and treatment guidance.

Recent studies have revolutionized the classification and treatment of MN as well. Rather than a single disease, MN represents a pattern of injury caused by the interaction of autoantibodies against podocitary antigens. The best therapeutic approach should target the pathogenic mechanism responsible. In this setting, the use of new generation technologies, such as MS or single cell RNA sequencing, and omics approaches could allow the better assessment of autoantibody profiling and disease prognosis in an unbiased fashion and may be helpful for a precision medicine approach towards patient management.

## Figures and Tables

**Figure 1 ijms-24-13134-f001:**
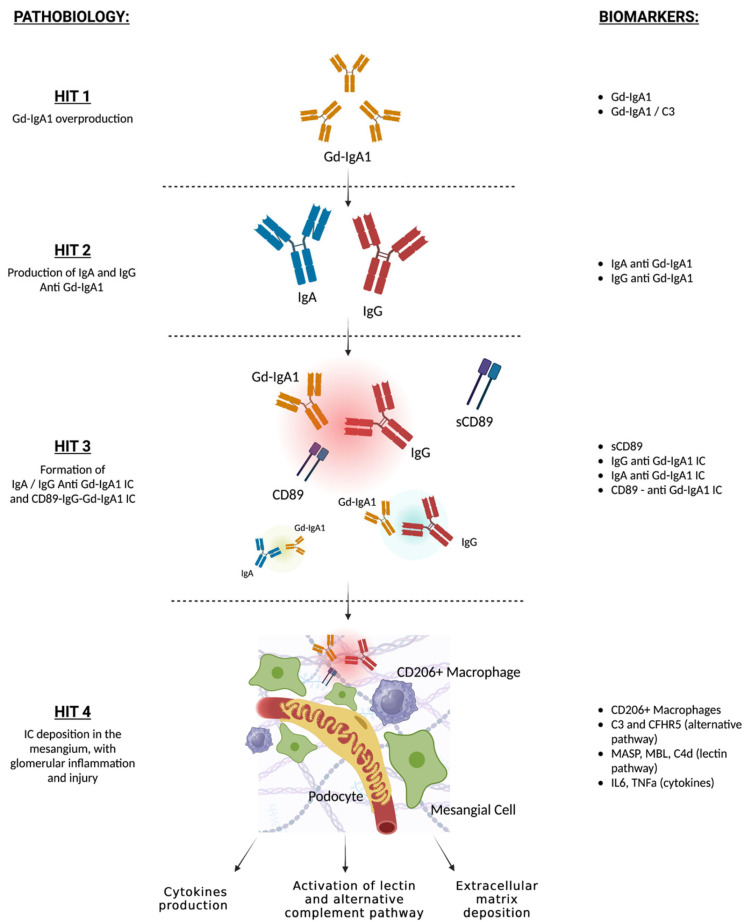
Four-hit model of IgAN pathogenesis and related potential biomarkers. Aberrant production of Gd-IgA1 (HIT 1). Increased circulant Gd-IgA1 and Gd-IgA1/C3 ratio are predictive of IgAN diagnosis and worse prognosis; autoantibodies formation against Gd-IgA1 (HIT 2). Higher levels of IgG or IgA anti-Gd-IgA1 are associated with IgAN; formation of IgG or IgA anti-Gd-IgA1 IC (HIT 3), which may bind sCD89. Higher levels of IgG or IgA anti-Gd-IgA1 IC are associated with IgAN, low serum and urinary CD89-IgA complexes are associated with active IgAN and worse prognosis; deposition of immune complexes in the mesangium, and inflammation and activation of lectin and alternate complement pathways (HIT 4). Glomerular CD106+ macrophage infiltration is a marker of immune activity, circulating levels and glomerular deposition of C3, CFHR5, MASP, MBL, and C4d are associated with proteinuria, active glomerular lesions, and worse prognosis. Gd-IgA1: galactose deficient IgA1; IC: immune complexes; sCD89: soluble CD89.

## Data Availability

Not applicable.

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
