# Peer review of "Present and Future of IgA Nephropathy and Membranous Nephropathy Immune Monitoring: Insights from Molecular Studies"

_ijms, 2023, doi:10.3390/ijms241713134_

Round 1

Reviewer 1 Report

Zanoni et al. present the immunologic perspectives for IgA nephropathy and membranous nephropathy in present and future diagnostic and therapeutic biomarkers. Their review is thoroughly in-depth.

 1. Authors summarize the clinically potential biomarkers for diagnosis and therapeutic prognosis in IgA nephropathy and membranous nephropathy. However, if possible, authors need to more detailed clinical meaning in immune response for diagnosis and therapeutics.

2. Comparing current KDIGO guidelines, what is the clinical meaning of testing novel biomarkers? Especially, IgA nephropathy has heterogeneous pathophysiology. Therefore, immune biomarkers might have less clinical meaning instead of current guidelines. Authors need to discuss the more detailed clinical meaning of immune biomarkers.

Author Response

Zanoni et al. present the immunologic perspectives for IgA nephropathy and membranous nephropathy in present and future diagnostic and therapeutic biomarkers. Their review is thoroughly in-depth.

  1. Authors summarize the clinically potential biomarkers for diagnosis and therapeutic prognosis in IgA nephropathy and membranous nephropathy. However, if possible, authors need to more detailed clinical meaning in immune response for diagnosis and therapeutics.

We thank the reviewer for this important suggestion. We have added a small summary at the end of each section that illustrates the clinical meaning of the discussed biomarkers.

  1. Comparing current KDIGO guidelines, what is the clinical meaning of testing novel biomarkers? Especially, IgA nephropathy has heterogeneous pathophysiology. Therefore, immune biomarkers might have less clinical meaning instead of current guidelines. Authors need to discuss the more detailed clinical meaning of immune biomarkers.

We thank the reviewer for this comment. We agree that, given the highly heterogeneous pathogenesis of IgAN, it is unlikely that a single immune biomarker will be useful for prognosis assessment and therapeutic guidance, as we have highlighted in the text. Current guidelines do not identify biomarkers for therapeutic guidance or specific immune activity, given that eGFR, proteinuria, the Oxford classification and the IgAN prognostic tool are not strictly correlated with IgAN immunological activity, but have a more general role on prognostic assessment. In this reviewed manuscript, we have explained that in better detail.

We really would like to thank you for your comments and suggestions, and we believe that they have significantly helped improve the manuscript.

Reviewer 2 Report

In the manuscript: “Present and future of IgA Nephropathy and Membranous Nephropathy immune monitoring: insights from molecular studies”, the authors reviewed IgA Nephropathy (IgAN) and Membranous Nephropathy (MN) as primary immune mediated glomerular diseases. Authors also reviewed the current evidence and potential novel biomarkers of immune activity based in IgAN and MN pathogenesis. An interesting topic that contributes to knowledge in the area, but certain issues must be corrected.

Major revisions

1.    In the abstract and introduction, the authors must mention what is the gap that the manuscript will fill within the current knowledge, the results that readers will found in the manuscript, and the possible conclusions that the reader will find through the manuscript.

2.    At the end of each sections the results must have concluded. That is, authors must mention the conclusions of each result: for instance, these results together suggest… or we concluded that…

Minor revisions

1.    Define all abbreviations to be presented in the text (for example: TNF and IL in lines 109).

2.    Please do not repeat definitions (line 28 and line 60)

No comments

Author Response

In the manuscript: “Present and future of IgA Nephropathy and Membranous Nephropathy immune monitoring: insights from molecular studies”, the authors reviewed IgA Nephropathy (IgAN) and Membranous Nephropathy (MN) as primary immune mediated glomerular diseases. Authors also reviewed the current evidence and potential novel biomarkers of immune activity based in IgAN and MN pathogenesis. An interesting topic that contributes to knowledge in the area, but certain issues must be corrected.

Major revisions

  1. In the abstract and introduction, the authors must mention what is the gap that the manuscript will fill within the current knowledge, the results that readers will found in the manuscript, and the possible conclusions that the reader will find through the manuscript.

We would like to thank the reviewer for this helpful suggestion. We have edited the abstract and the introduction with a more detailed description of our review purpose.

  1. At the end of each sections the results must have concluded. That is, authors must mention the conclusions of each result: for instance, these results together suggest… or we concluded that…

We thank you for this suggestion. At the end of each section, we have added a brief summary. We believe that your comments and recommendations have significantly helped improve the manuscript.

Minor revisions

  1. 1. Define all abbreviations to be presented in the text (for example: TNF and IL in lines 109).
  2. Please do not repeat definitions (line 28 and line 60)

We thank the reviewer for this suggestion. We have added the definitions and deleted the repeated ones.

Reviewer 3 Report

no commments

Author Response

We thank the reviewer for reading through our manuscript, and hope that you will consider it for publication in its reviewed form.

Reviewer 4 Report

The review, "Present and Future of IgA Nephropathy and Membranous Nephropathy Immune Monitoring: Insights from Molecular Studies," reviews the advances in understanding the pathogenesis of IgA nephropathy (IgAN) and membranous nephropathy (MN) through the use of molecular technologies.

 The authors highlight the significant progress made in recent years thanks to the development of molecular techniques, such as whole-genome genotyping, RNA sequencing, and mass spectrometry. These technologies have contributed to a better understanding of the underlying molecular mechanisms and pathways involved in IgAN and MN. The review also emphasized the importance of an individualized approach to treatment based on the assessment of immune activity and prognosis. The authors discuss how recent genome-wide association studies have identified potential targets for immunomodulatory agents, offering new avenues for therapeutic interventions.

 However, this work still needs some refinement.

As for the prepared manuscript, it does not comply with the formatting for IJMS, which probably indicates the rejection of this manuscript by other publishers. Links to testimonials observed in the article need to be formatted correctly; the same goes for testimonials.

 In addition, when highlighting subsections, the sprinkle character is unedited.

   Another area for improvement is the inappropriate formatting of the table. In the context of the table, I also propose to include a column with references so that the reader, reading the description of the biomarker, could refer to the publication that mentions such a function.

 As for the Figure, in my opinion, it should be enlarged, then it will gain legibility.

 Another issue is the methodology of preparing the literature review. How the authors performed this literature review, and what protocol they used? What words did they use to search for particular thematic articles, and what were the criteria for inclusion and exclusion of publications?

For my part, I suggest following the PRISMA protocol so that the reader knows to what extent the authors searched for specific literature.

 In my opinion, the last point is that the authors should expand the subsections related to genomics.

 Finally, he would like to ask if the authors could point to aspects of their work that distinguish it from other similar studies that can be found for particular diseases.

 To sum up, after receiving a response to the review and considering the corrections, I think you can consider publishing the article in IJMS.

Author Response

The review, "Present and Future of IgA Nephropathy and Membranous Nephropathy Immune Monitoring: Insights from Molecular Studies," reviews the advances in understanding the pathogenesis of IgA nephropathy (IgAN) and membranous nephropathy (MN) through the use of molecular technologies.

 The authors highlight the significant progress made in recent years thanks to the development of molecular techniques, such as whole-genome genotyping, RNA sequencing, and mass spectrometry. These technologies have contributed to a better understanding of the underlying molecular mechanisms and pathways involved in IgAN and MN. The review also emphasized the importance of an individualized approach to treatment based on the assessment of immune activity and prognosis. The authors discuss how recent genome-wide association studies have identified potential targets for immunomodulatory agents, offering new avenues for therapeutic interventions.

 However, this work still needs some refinement.

As for the prepared manuscript, it does not comply with the formatting for IJMS, which probably indicates the rejection of this manuscript by other publishers. Links to testimonials observed in the article need to be formatted correctly; the same goes for testimonials.

 In addition, when highlighting subsections, the sprinkle character is unedited.

We would like to thank the reviewer for this suggestion. We have reformatted the manuscript accordingly.

  Another area for improvement is the inappropriate formatting of the table. In the context of the table, I also propose to include a column with references so that the reader, reading the description of the biomarker, could refer to the publication that mentions such a function.

 As for the Figure, in my opinion, it should be enlarged, then it will gain legibility.

We thank you for pointing out this issue. We have reformatted the tables and enlarged the figure. We have also moved the references within the table to a separate, additional column.

Another issue is the methodology of preparing the literature review. How the authors performed this literature review, and what protocol they used? What words did they use to search for particular thematic articles, and what were the criteria for inclusion and exclusion of publications?

For my part, I suggest following the PRISMA protocol so that the reader knows to what extent the authors searched for specific literature.

 We would like to thank the reviewer for this important comment. We would like to state that our work is not a systematic review, but a narrative review, where we focused our search on available and novel biomarkers based on current understanding of disease pathogenesis. We mainly focused our search on biomarkers that are related to disease pathogenic mechanisms, and we discussed the use of omics in patient risk stratification and clinical guidance. Therefore, our literature search has been performed accordingly by using specific key words related to each section (i.e. biomarkers, antibodies, complement, immune monitoring, immune cells, genetic risk, RNA sequencing, transcriptomics, proteomics..). We selected the studies that focused on immune activity markers and evidence of their correlation with prognosis and therapeutic response. However, we did not strictly follow the PRISMA protocol, since we aimed to create a narrative review manuscript rather than a systematic review.

Nevertheless, we believe that our work could be helpful for guidance for future research studies, and we do hope that you would consider its revised version for publication.

In my opinion, the last point is that the authors should expand the subsections related to genomics.

We thank the reviewer for this suggestion. We have extended the discussion in the genomic subsection of both IgAN and MN. 

Finally, he would like to ask if the authors could point to aspects of their work that distinguish it from other similar studies that can be found for particular diseases.

We thank the reviewer for this request. Our manuscript is a narrative review that focuses on the progression that has been made over the last few years in the search for new biomarkers of immune activity. We have included a thorough discussion on molecules and cell subpopulations, as well as results from more recent approaches, such as genomics, transcriptomics/epigenomics and proteomics. Therefore, we have included very recent studies using omics techniques both diseases, and illustrated the potential use of molecular profiling, a future approach, to risk stratify and manage individuals affected by IgAN and MN. In conclusion, although many review manuscripts have been published in the past on this matter, our work adds a particular focus on molecular findings and omics, and discusses very recent studies. We ultimately hope that our review manuscript, if considered for publication, would be able to guide future research efforts on molecular biomarkers and clinical trial design.

 To sum up, after receiving a response to the review and considering the corrections, I think you can consider publishing the article in IJMS.

Round 2

Reviewer 4 Report

I appreciate the authors' responsiveness in making corrections to their manuscript. While I agree that the authors have prepared a narrative review, I would like to request the addition of information about their literature search strategy if they choose not to rely on the Gradient PRISMA protocol. Specifically, I would like to see information about the keywords, years, inclusion and exclusion criteria, and other search parameters used, much like what is done with patients. Given the vast amount of literature available, finding specific information can be a challenge. Providing this information would enable interested readers to conduct similar searches and quickly narrow down the literature to their specific topic of interest. In my opinion, adding a materials and methods section that addresses these concerns would improve the effectiveness of the authors' work.

Author Response

We thank the reviewer for this suggestion. We decided to add a supplementary description of our methods for studies selection.